# Increased risk of adverse events in non-cancer patients with chronic and high-dose opioid use—A health insurance claims analysis

Jakob M. Burgstaller[1,2☯‡], Ulrike Held[1,3☯‡], Andri Signorell[4], Eva Blozik[2,4], Johann Steurer[1], Maria M. Wertli[1,5]*

1 Department of Internal Medicine, Horten Center for Patient Oriented Research and Knowledge Transfer, University of Zurich, Zurich, Switzerland, 2 Institute of Primary Care, University and University Hospital Zürich, Zürich, Switzerland, 3 Department of Biostatistics at Epidemiology, Biostatistics and Prevention Institute, University of Zürich, Zurich, Switzerland, 4 Department of Health Sciences, Helsana, Dübendorf, Switzerland, 5 Department of General Internal Medicine, Bern University Hospital, University of Bern, Bern, Switzerland

☯ These authors contributed equally to this work.
‡ These authors share first authorship on this work.
* Maria.Wertli@insel.ch

**Data Availability Statement:** Data underlying the study belongs to a third party and cannot be made publicly available. This study is based on administrative, de-identified insurance claims data

## Abstract

### Background

Chronic and high dose opioid use may result in adverse events. We analyzed the risk associated with chronic and high dose opioid prescription in a Swiss population.

### Methods

Using insurance claims data covering one-sixth of the Swiss population, we analyzed recurrent opioid prescriptions (≥2 opioid claims with at least 1 strong opioid claim) between 2006 and 2014. We calculated the cumulative dose in milligrams morphine equivalents (MED) and treatment duration. Excluded were single opioid claims, opioid use that was cancer treatment related, and opioid use in substitution programs. We assessed the association between the duration of opioid use, prescribed opioid dose, and benzodiazepine use with emergency department (ED) visits, urogenital and pulmonary infections, acute care hospitalization, and death at the end of the episode.

### Results

In 63,642 recurrent opioid prescription episodes (acute 38%, subacute 7%, chronic 25.8%, very chronic (>360 days) episodes 29%) 18,336 ED visits, 30,209 infections, 19,375 hospitalizations, and 9,662 deaths occurred. The maximum daily MED dose was <20 mg in 15.8%, 20−<50 mg in 16.6%, 50−<100 mg in 21.6%, and ≥100 mg in 46%. Compared to acute episodes (<90 days), episode duration was an independent predictor of ED visits (chronic OR 1.09 (95% CI 1.03–1.15), very chronic (>360 days) OR 1.76 (1.67–1.86)) for adverse effects; infections (chronic OR 1.74 (1.66–1.82), very chronic 4.16 (3.95–4.37)),

handled in compliance with privacy law and regulations. The analyses were performed within the premises of the Department of Health Sciences at the Helsana and only aggregated data was shared. Aggregated data can be made available only on reasonable request after assessment and permission of the data owner (Helsana Insurances Group). Data requests can be made to: Department of Health Sciences, Helsana, Zürichstrasse 130, CH-8600 Dübendorf, Switzerland (website: https://www.helsana.ch/de/private, phone: +41 58 340 18 80, email: Gesundheitskompetenz@helsana.ch). The authors confirm that they did not have any special access to this data which other researchers would not have.

**Funding:** The study was not funded. The insurance company, Helsana Group, providing the administrative data did not have an influence on the study design, the interpretation of the data, the writing of the report, and in the decision to submit the paper for publication.

**Competing interests:** MMW, JMB, JS, and UH have no competing interest to declare. AS and EB are employed by the Helsana Group. The sponsor had no role in the planning the study, conducting the analysis, or submission of this manuscript. This does not alter our adherence to PLOS ONE policies on sharing data and materials. Helsana Group shall have no liability to any third party in respect to the contents of this article.

and hospitalization (chronic: OR 1.22 (1.16–1.29), very chronic OR 1.82 (1.73–1.93)). The risk of death decreased over time (very chronic OR 0.46 (0.43–0.50)). A dose dependent increased risk was observed for ED visits, hospitalization, and death ($\geq$100mg daily MED OR 1.21 (1.13–1.29), OR 1.29 (1.21–1.38), and OR 1.67, 1.50–1.85, respectively). A concomitant use of benzodiazepines increased the odds for ED visits by 46% (OR 1.46, 1.41–1.52), infections by 44% (OR 1.44, 1.41–1.52), hospitalization by 12% (OR 1.12, 1.07–1.1), and death by 45% (OR 1.45, 1.37–1.53).

## Conclusion

The length of opioid use and higher prescribed morphine equivalent dose were independently associated with an increased risk for ED visits and hospitalizations. The risk for infections, ED visits, hospitalizations, and death also increased with concomitant benzodiazepine use.

## Background

Chronic pain is a leading cause for years lived with disability worldwide [1]. Effective pain management is needed to decrease pain-associated disability and improve the quality of life. Since the 1990ies the World Health Organizations' (WHO) pain relief ladder [2, 3] has been used to improve pain control in cancer and non-cancer patients. A stepwise increase of treatment intensity in cancer pain is recommended with non-opioids being the first choice for mild pain. Weak opioids (e.g., tramadol, codeine) are recommended for mild to moderate pain, and for severe pain, strong opioids (e.g., morphine, fentanyl) [3]. Advocacy for pain control, advertisement of efficacy of opioids for chronic pain based on low level evidence, and aggressive prescription practices by physicians resulted in an increased use of opioids for cancer and non-cancer related pain [4–6].

Whereas opioids are well established for the relief of acute severe pain in patients with active cancer, strong opioids are no more effective in chronic non-cancer pain than non-opioid medications [7–9]. Due to the potential adverse events [9–11], opioids are considered second line drugs [7–9]. Despite these recommendations, opioids (particularly strong opioids) are increasingly prescribed for chronic non-cancer pain and the use of strong opioids has reached enormous dimensions in some countries [12–14]. In parallel to a steep increase in the opioid use in the US [7], an increase in unintentional opioid overdose and higher hospital admission rates has been observed [15–19]. In Europe, only limited information on the impact of chronic opioid use is available. Studies in Europe also showed an increased use of strong opioids [20–24], but the consequences are less well described in the literature. A study using a clinical database for primary care practices in the UK reported an increased risk for serious adverse events such as major trauma, addiction and overdose in chronic opioid use [25]. In Switzerland, the use of strong opioids more than doubled between 2006 and 2013 [26]. The implications of chronic opioid use for non-cancer pain remain unknown.

The aim of this study was to assess the risk of adverse events in recurrent opioid use for non-cancer pain in a representative sample of the Swiss population. We hypothesized that long-term opioid use and higher opioid doses are associated with a higher rate of adverse events.

## Methods

### Sources of data

In Switzerland, compulsory basic health insurance is universal and covers the population of 8.2 million persons with a comprehensive benefits package defined by federal authorities [27]. The study cohort was identified from the computerized insurance claims database from one of the major health insurers in Switzerland. The Helsana insurance group covers 1.2 million individuals (approximately one-sixth of the Swiss population) in all 26 administrative regions (cantons). The patient-level linked database provides information on socio-demographic data, health insurance status, prescribed drugs, health care encounters for pharmacy, hospital, outpatient, and nursing home services. In case of death, the date of death is also included in the database. The study population was limited to beneficiaries with full compulsory insurance coverage during the observation period. All persons ever involved in a drug substitution program were excluded.

The Swiss health care system is highly decentralized and no centralized opioid registry is available. Opioids cannot be purchased over the counter and for strong opioids, a special prescription with a unique identification number (a so called "prescription for narcotic substances") is issued with 3 copies. One copy remains with the prescribing physician, one with the pharmacy, and one with the insurance company. Although the regulation minimizes the risk of abuse, potential misuse (e.g. multiple prescribers) cannot be identified because no central database exists. Further, pharmacies send their bills directly to the insurance company. Therefore, opioid claims cover close to 100% of all prescribed opioids.

### Study cohort

The study cohort included adult patients (aged 18 years and older) with recurrent opioid claims between January 2006 and December 2014 were included [28]. Recurrent opioid use was defined as ≥2 consecutive opioid claims including at least 1 strong opioid claim. We identified opioid claims using unique codes for the pharmaceutical class based on the WHO pharmacological Anatomical Therapeutic Chemical (ATC) classification system (S1 Table [29]). Opioids were defined as weak in opioid formulations with a morphine conversion factor of 0.3 or less (N02AA59 (codeine and combinations), N02AX01 (tilidine), N02AX02 (tramadol), and N02AX06 (tapentadol)). All opioids with a morphine conversion factor of >0.3 were defined as strong.

We excluded opioid use related to cancer treatment. Cancer-related use was defined when a cancer specific treatment occurred (≥1 pre-specified ATC or outpatient Swiss tariff positions (Tarmed) codes, S2 Table) within three months before and after the first filled opioid prescription. As insurance companies reimburse opioid use in substitution programs since 1999, we identified cases using specific reimbursement codes (Tarmed Position 00.0155, positions specifically assigned to substitution programs in pharmacies or substitution centers for buprenorphine, methadone, heroin, and morphine). All people were excluded when a substitution code was identified (e.g., in a patient, the unique code was identified in the database in 2009 then all opioids reimbursed for this person were excluded). In addition, we excluded diamorphine using the corresponding ATC-code (N07BC06 Diaphin®). Other specific brands are used within substitution programs and for pain treatment: Sevre-Long® (morphine, N02AA01), Subutex® (N07BC01) and Temgesic® (N02AE01, both buprenorphine), or L-Polamidon® (N07BC05) and Ketalgin® (N07BC02, both methadone). These medications were included in the analysis as long as no code for a substitution program was detected. We excluded patients with recurrent prescription of Subutex® sublingual (not excluded by the above-defined

criteria), when the daily dose was more than 640mg morphine equivalent assuming that Subutex® was used within an "off-label" opioid substitution.

## Medication exposure

An episode of opioid treatment began on the day a patient filled the first opioid prescription. The duration of an opioid episode was calculated using the difference (in days) between the date the initial prescription was dispensed and the run-out date of the last prescription dispensed plus 1 [30]. In case of several claims, the time between the last two claims and the calculated average daily dose (see below) were used to calculate the run-out date. An opioid episode ended when three months after the calculated run-out date, no new claim was filled. We considered the follow-up period of 7 days after the first opioid claim until the end of the episode for all endpoints. Episodes lasting beyond December 2014 were censored by December 31, 2014.

## Definitions

We used the following definitions proposed by von Korff et al [30]:

**Morphine equivalents per episode.** MED for each prescription dispensed during the episode. Each reimbursement of an opioid medication (referred to hereafter as a "claim") was converted to morphine equivalent dose (MED) as follows: Strength of opioid drug in mg per unit *x* quantity of units per reimbursed package *x* number of packages *x* conversion factor for morphine equivalents. The equianalgesic dose conversions are estimates and cannot account for individual variability in genetics and pharmacokinetics. Wherever available we used conversion factors provided by the Swiss Agency for Therapeutic Products (Swissmedic, agency comparable to the US Food and Drug Administration, FDA) or the morphine equivalent conversion factor per mg of opioid was based on the CONSORT classification (CONsortium to Study Opioid Risks and Trends [30]). Further, we consulted the literature relevant to the topic and a clinical pharmacologist (See S1 Table: opioids, examples of brand names, the morphine equivalent conversion factors, and the route of administration).

The MED calculation for patches was based on the assumption that one patch delivers opioids over a time provided by the manufacturer. For example, fentanyl patches deliver the dispensed (and bioavailable) mcg per hour over 72 hours. The calculation of the total bioavailable MED dose in mg equals (mcg/hour (according to the package reimbursed) × 72 hours' × number of patches per package × number of packages reimbursed × 100 [fentanyl conversion factor]) / 1000. The total MED in mg for one package containing 10 fentanyl patches that each delivers 12mcg per hour is calculated as follows: 12mcg × 72h × 10 patches × 100 = 864,000mcg/1000 = 864mg. For transdermal buprenorphine patches the assumption is that one patch delivers the dispensed (and bioavailable) mcg per hour over 96 hours. The total MED dose in milligram equals (mcg/h according to the package reimbursed × 96 hours' × number of patches per package × number of packages reimbursed × 95 [buprenorphine conversion factor]) / 1000.

**Duration of episode.** Episodes were categorized by their duration into acute (<90 days), subacute (≥90 to <120 days or <10 claims), chronic (≥90 days and ≥10 claims or ≥120 days' supply of opioids, and very chronic use (>360 days)) [30].

**Average daily dose.** Total MED for an episode divided by the episode duration (days). In case of several claims, the MED per treatment day was calculated between two claims and categorized into one of the four groups: <20; ≥20 to <50; ≥50 to <100, and ≥100mg MED per day.

**Costs per day.** We divided the sum of all reimbursed claims and treatment costs (outpatient and inpatient costs) by the episode duration in days.

## Outcome of interest

As primary outcomes, we examined emergency department visits, the occurrence of urogenital and pulmonary infections, acute care hospitalization, and death during the episode. According to the definition of the WHO, adverse events are medical occurrences temporally associated with the use of a medical product, but not necessarily causally related. Adverse events were included in the analysis when they occurred two weeks or later after the index date of the opioid prescription. Antibiotic medications typically prescribed for urogenital and pulmonary infections were identified by ACT codes (J01MA02 ciprofloxacin, J01MA06 norfloxacin, J01EE01 sulfamethoxazole and trimethoprim, J01XX01 fosfomycin) and pulmonary infections (J01AA02 doxycycline, J01CR02 amoxicillin and enzyme inhibitor, J01CA04 amoxicillin, J01FA09 clarithromycin, J01MA14 moxifloxacin). The date of death was available within the database. No cause of death was available for the current analysis. Because no information on diagnoses and misuse were available in the insurance claims database, we were not able to assess these two outcomes.

## Confounders

Because the risk of ED visits, infections, hospitalization, and death can influence use of opioids and the choice of the pharmacologic agents, the analysis controlled for 22 confounders potentially associated with opioid use and the outcomes. Confounders included demographic information (age, sex), cultural factors (language region of residence), insurance type (additional private insurance, managed care models), concomitant benzodiazepine use, and comorbid diseases. Comorbid diseases were based on an adapted version of the Chronic Disease Score (CDS [31, 32]) and categorized into chronic infections, inflammatory disease, renal disease, endocrine disease, diabetes, pulmonary diseases, liver failure, organ transplant, neurologic disease, cardiac disease, hyperlipidemia, glaucoma, acid peptic disease, thyroid disease, and gout (details of the codes are provided in S3 Table). The CDS has been shown to be associated with health care utilization [31, 33]. Additionally, we included the pharmacological agent of the strong opioid (i.e. morphine, oxycodone, buprenorphine, fentanyl, hydromorphone, and pethidine) as a proxy for the complexity of an episode (e.g. opioid rotation with changes of substances during an episode or preferred use of a substance depending on the clinical situation). As we did not have information on the indication for opioid prescription and pain intensity, changes of pharmacological agents within one episode may indicate more complex pain problems.

## Statistical methods

Descriptive statistics included median and interquartile range for the continuous parameters, and percentages for the categorical outcomes. We compared groups using Kruskal-Wallis test, Fisher's exact test, and Chi-square tests wherever appropriate. We fitted logistic regression models to the binary outcomes of interest including disease duration (acute, subacute, chronic, very chronic), maximum prescribed dose (<20 mg (reference), 20 - <50 mg, 50 - <100mg, ≥100 mg, prescribed active morphine compounds. The following potential relevant confounders were included in the models: age, sex, additional insurance models (i.e. private insurance policies, managed care models), place of residency (living in an Italian/French or German speaking canton), chronic infections, chronic inflammatory disease, diabetes, cardiac disease, renal disease, end stage renal disease, gout, liver failure, organ transplant, thyroid disease, and

neurologic disease. Results are reported in odds ratio (OR) including the 95% confidence interval (95% CI). Overall treatment costs were calculated based on all reimbursed in- and out-patient costs. We calculated the percent increase or decrease in treatment costs using loglinear model. Results are reported in % in- or decrease including the 95% CI. Statistical analysis was performed using the statistical programming language R (https://www.r-project.org/) [34]. The following packages were used: DescTools, mvtnorm, foreign, Rcpp, RDCOMClient, and tcltk.

### Ethics statement

This study is based on administrative de-identified insurance claims data handled in compliance with privacy law and regulations. According to the local ethical committee (Ethical committee of the Canton Zurich, Switzerland) no IRB approval was required. The study was conducted following the principles of good clinical practice and in accordance with the Declaration of Helsinki.

## Results

Out of 591,633 opioid claims, 76,968 recurrent opioid claims were identified. We excluded 13,326 episodes because they were related to a cancer treatment, leaving 63,642 recurrent opioid prescriptions episodes for non-cancer indications for further analysis. The episode duration was in 38% acute, 7% subacute, 25.8% chronic, and 29% very chronic (Table 1). The median age of patients was 72 years (IQR 56; 82), the majority were female (65.5%), and 19.0% of the patients lived in a French or Italian speaking part of Switzerland. The maximum daily dose was <20 mg in 15.8%, 20−<50 mg in 16.6%, 50−<100 mg in 21.6%, and ≥100 mg in 46%. The maximum daily dose of ≥100 mg was used in 34.6% of the acute, 29.2% of the subacute, 41.4% of the chronic, and in 69% of very chronic episodes (>360 days).

The most prevalent treatments observed were for cardiac diseases (60.8%), psychiatric conditions (42.2%), chronic inflammatory diseases (37.7), pulmonary diseases (14.3%), for chronic infections (10.5%), and diabetes (12.1%). Compared to the acute opioid episodes, the proportions increased in very chronic episodes for cardiac diseases (40.3 to 81.2%), psychiatric diseases (21.4 to 66.9%), inflammatory diseases (21.3 to 81.2%), pulmonary diseases (6.6 to 24.7%), chronic infections (5.3 to 18.5%), and diabetes (7.0 to 17.2%).

### Outcomes of interest

Overall, 18,336 ED visits, 30,209 infections, 19,375 hospitalizations, and 9,662 deaths occurred during the opioid episodes. ED visits occurred in 26.6% in acute episodes (n = 4,880), in 5.2% in subacute (n = 951), in 23.6% during chronic (n = 4,323), and in 44.6% in very chronic episodes (n = 8,182, Table 2). Infections occurred in 22.3% during acute (n = 6,737), in 5.5% during subacute (n = 1,669), in 26.3% in chronic (n = 7,948), and in 45.9% in very chronic (3,855) episodes. Hospitalizations were in 25% during acute (n = 4,852), in 6.1% during subacute (n = 1,176), in 25.5% during chronic (n = 4,938), and in 43.4% during very chronic (n = 8,409) episodes. The majority of deaths occurred during an acute episode (n = 4,309, 44.6%). Deaths occurred in 5.8% during subacute (n = 562), in 22.5% during chronic (n = 2,173), and in 27.1% in very chronic (n = 2,618).

### Risk for ED visits

ED visits occurred in 11.3% in episodes with a daily dose of <20mg compared to 56.4% in episodes with a daily dose of ≥100mg (Table 2). After adjustment for potential confounders, we

**Table 1. Baseline characteristics.**

| | Total | Acute | Subacute | Chronic | Very chronic | p-value |
|---|---|---|---|---|---|---|
| | N (%) / median [IQR] / mean (SD) | | | | | |
| Number | 63'642 (100) | 24'220 (38.1) | 4'446 (7.0) | 16'427 (25.8) | 18'549 (29.1) | |
| Age | 72.0 [56; 82] | 72.0 [55; 82] | 71.0 [55; 81] | 72.0 [56; 81] | 73.0 [58, 82] | <0.000 |
| Female | 41'699 (65.5) | 14'856 (61.3) | 2'818 (63.4) | 10'740 (65.4) | 13'285 (71.6) | <0.000 |
| (Semi)private insurance | 13'089 (20.6) | 5'098 (21.0) | 911 (20.5) | 3'316 (20.2) | 3'764 (20.3) | n.s. |
| Managed care | 13'517 (21.2) | 6'030 (24.9) | 994 (22.4) | 3'459 (21.1) | 3'034 (16.4) | <0.000 |
| Italian / French part | 12'068 (19.0) | 4'230 (17.5) | 810 (18.2) | 3'261 (19.9) | 3'767 (20.3) | <0.000 |
| Daily dose category | | | | | | <0.000 |
| <20 mg | 10'054 (15.8) | 4'781 (19.7) | 1'159 (26.1) | 3'082 (18.8) | 1'032 (5.6) | |
| 20 - <50 mg | 10'552 (16.6) | 4'992 (20.6) | 904 (20.3) | 2'915 (17.7) | 1'741 (9.4) | |
| 50 - <100mg | 13'747 (21.6) | 6'061 (25.0) | 1'085 (24.4) | 3'633 (22.1) | 2'968 (16.0) | |
| ≥100 mg | 29'289 (46.0) | 8'386 (34.6) | 1'298 (29.2) | 6'797 (41.4) | 12'808 (69.0) | |
| Episode Duration | 145 [51; 455] | 37 [18; 59] | 103 [96; 111] | 202 [154; 271] | 829 [541; 1406] | <0.000 |
| Benzodiazepine | 21'548 (33.9) | 4'634 (19.1) | 1'219 (27.4) | 5'898 (35.9) | 9'797 (52.8) | <0.000 |
| Substances | | | | | | |
| Morphine | 20'934 (32.9) | 8'154 (33.7) | 1'207 (27.1) | 4'927 (30.0) | 6'646 (35.8) | <0.000 |
| Oxycodone_ | 25'054 (39.4) | 8'398 (34.7) | 1'679 (37.8) | 6'688 (40.7) | 8'289 (44.7) | <0.000 |
| Fentanyl | 20'832 (32.7) | 6'373 (26.3) | 1'261 (28.4) | 5'369 (32.7) | 7'829 (42.2) | <0.000 |
| Pethidine | 7'400 (11.6) | 3'368 (13.9) | 640 (14.4) | 1'738 (10.6) | 1'654 (8.9) | <0.000 |
| Buprenorphine | 5'798 (9.1) | 1'395 (5.8) | 314 (7.1) | 1'570 (9.6) | 2'519 (13.6) | <0.000 |
| Hydromorphone | 2'227 (3.5) | 498 (2.1) | 118 (2.7) | 536 (3.3) | 1'075 (5.8) | <0.000 |
| Comorbidities | | | | | | |
| Chronic infections | 6'690 (10.5) | 1'276 (5.3) | 342 (7.7) | 1'639 (10.0) | 3'433 (18.5) | <0.000 |
| Chronic inflammatory disease | 23'990 (37.7) | 5'165 (21.3) | 1'373 (30.9) | 6'587 (40.1) | 10'865 (58.6) | <0.000 |
| Renal disease | 702 (1.1) | 120 (0.5) | 45 (1.0) | 195 (1.2) | 342 (1.8) | <0.000 |
| End stage renal disease | 520 (0.8) | 112 (0.5) | 24 (0.5) | 155 (0.9) | 229 (1.2) | <0.000 |
| Diabetes | 7'696 (12.1) | 1'706 (7.0) | 570 (12.8) | 2'233 (13.6) | 3'187 (17.2) | <0.000 |
| Pulmonary disease | 9'080 (14.3) | 1'592 (6.6) | 506 (11.4) | 2'394 (14.6) | 4'588 (24.7) | <0.000 |
| Liver failure | 5'913 (9.3) | 1'102 (4.5) | 297 (6.7) | 1'426 (8.7) | 3'088 (16.6) | <0.000 |
| Organ transplant | 650 (1.0) | 134 (0.6) | 38 (0.9) | 170 (1.0) | 308 (1.7) | <0.000 |
| Neurologic disease | 3'775 (5.9) | 584 (2.4) | 177 (4.0) | 921 (5.6) | 2'093 (11.3) | <0.000 |
| Cardiac disease | 38'704 (60.8) | 9'759 (40.3) | 2'713 (61.0) | 11'164 (68.0) | 15'068 (81.2) | <0.000 |
| Thyroid disease | 4'929 (7.7) | 900 (3.7) | 315 (7.1) | 1'464 (8.9) | 2'250 (12.1) | <0.000 |
| Gout | 2'531 (4.0) | 395 (1.6) | 148 (3.3) | 706 (4.3) | 1'282 (6.9) | <0.000 |
| Psychiatric disease | 26'837 (42.2) | 5'171 (21.4) | 1'594 (35.9) | 7'664 (46.7) | 12'408 (66.9) | <0.000 |
| Treatment costs per day# | 56.0 [27.5; 119.9] | 100.2 [50.0; 223.4] | 52.9 [28.3; 113, 4] | 43.1 [23.3; 94.2] | 34.7 [20.2; 65.6] | <0.000 |

#treatment costs per day (in Swiss Francs): All reimbursed in and outpatient treatment costs.

found an increased risk for ED visits with an increased duration of opioid use in chronic episodes (OR 1.09, 95% CI 1.03–1.15, Table 3, Fig 1), very chronic episodes (OR 1.76, 1.67–1.86), and in opioid episodes with doses ≥100mg daily MED (OR 1.21, 1.13–1.29). Concomitant use of benzodiazepines increased the odds by 46% (OR 1.46, 1.41–1.52).

## Risk for infections

Infections requiring antibiotic use occurred in 13.5% in episodes of <20mg compared to 53.2% in episodes with a daily dose of ≥100mg (Table 2). The duration of opioid episode was

**Table 2. Summary of adverse events.**

| | ED visits (N = 18,336) | Infections (N = 30,209) | Hospitalization (N = 19,375) | Deaths (N = 9,662) |
|---|---|---|---|---|
| | | N (%) | | |
| Acute episodes | 4,880 (26.6) | 6,737 (22.3) | 4,852 (25.0) | 4,309 (44.6) |
| Subacute | 951 (5.2) | 1,669 (5.5) | 1,176 (6.1) | 562 (5.8) |
| Chronic | 4,323 (23.6) | 7,948 (26.3) | 4,938 (25.5) | 2,173 (22.5) |
| Very chronic | 8,182 (44.6) | 3,855 (45.9) | 8,409 (43.4) | 2,618 (27.1) |
| Daily dose <20mg | 2'070 (11.3%) | 4'089 (13.5%) | 2'068 (10.7%) | 744 (7.7%) |
| 20 - <50mg | 2'419 (13.2%) | 4'181 (13.8%) | 2'796 (14.4%) | 1'105 (11.4%) |
| 50 - <100mg | 3'509 (19.1%) | 5'862 (19.4%) | 4'050 (20.9%) | 1'876 (19.4%) |
| ≥100mg | 10'338 (56.4%) | 16'077 (53.2%) | 10'461 (54.0%) | 5'937 (61.4%) |

independently associated with an increased risk for infections requiring antibiotic use (chronic OR 1.74 (1.66; 1.82), very chronic 4.16 (3.95; 4.37)) with no significant dose dependent effect. Co-prescribing of benzodiazepines increased the odds by 44% (OR 1.44, 1.41–1.52).

## Risk for hospitalization

Hospitalizations occurred in 10.7% in episodes of <20mg compared to 54% in episodes with a daily dose of ≥100mg (Table 2). An increasing duration of opioid use was associated with an increased chance for hospitalization (chronic: OR 1.22 (1.16–1.29), very chronic: OR 1.82 (1.73–1.93)). A dose dependent risk was observed (50–<100mg: OR 1.18 (1.10; 1.26), ≥100mg OR 1.29 (1.21; 1.38)). Co-prescribing of benzodiazepine increased the odds by 12% (OR 1.12, 1.07–1.16).

## Risk of death

Deaths occurred in episodes of <20mg in 7.7% compared to 61.4% in episodes with a daily dose of ≥100mg (Table 2). The majority of patients, died during an acute episode. With increasing episode duration, the risk of death decreased (chronic: OR 0.61, 0.57–0.66, very

**Table 3. Risk for adverse events.**

| | ED visits | Infections | Hospitalization | Death |
|---|---|---|---|---|
| | | OR (95% CI)§ | | |
| Duration (acute reference) | | | | |
| Subacute duration | 0.97 (0.89; 1.05) | **1.26 (1.17; 1.35)** | **1.19 (1.10; 1.29)** | **0.73 (0.66; 0.82)** |
| Chronic | **1.09 (1.03; 1.15)** | **1.74 (1.66; 1.82)** | **1.22 (1.16; 1.29)** | **0.61 (0.57; 0.66)** |
| very chronic (>360 days) | **1.76 (1.67; 1.86)** | **4.156 (3.95; 4.37)** | **1.82 (1.73; 1.93)** | **0.46 (0.43; 0.50)** |
| Daily dose (<20mg reference) | | | | |
| 20 - <50mg | 1.01 (0.94; 1.08) | 0.90 (0.85; 0.96) | **1.11 (1.03; 1.19)** | 1.04 (0.93; 1.16) |
| 50 - <100mg | 1.05 (0.98; 1.12) | 0.93 (0.87; 0.99) | **1.18 (1.10; 1.26)** | **1.19 (1.07; 1.33)** |
| ≥100mg | **1.21 (1.13; 1.29)** | 0.99 (0.94; 1.06) | **1.29 1.21; 1.38)** | **1.67 (1.50; 1.85)** |
| Co-prescription | | | | |
| Benzodiazepine | **1.46 (1.41; 1.52)** | **1.18 (1.14; 1.23)** | **1.12 (1.07; 1.16)** | **1.45 (1.37; 1.53)** |

ED, emergency department

§Adjusted for age, sex, chronic infections, chronic inflammatory disease, diabetes, cardiac disease, renal disease, end stage renal disease, gout, liver failure, organ transplant, thyroid disease, neurologic disease, (semi)private insurance status, living in an Italian/French speaking canton, managed care model, and pharmacological agents as a proxy for the complexity of the episode.

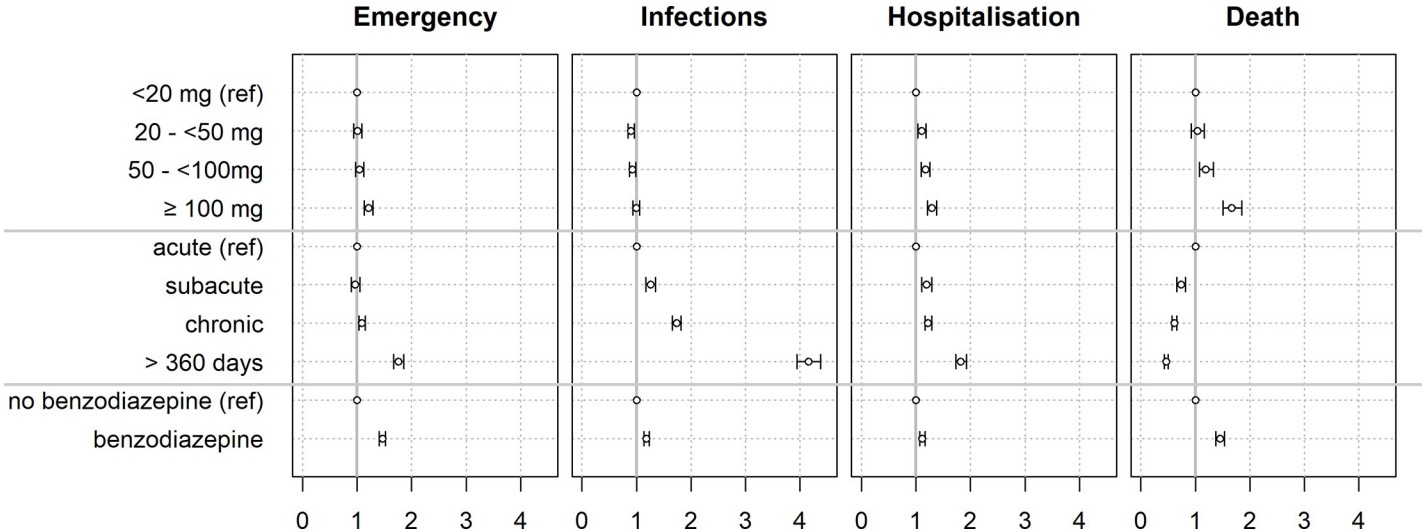

**Fig 1. Risk for adverse events during chronic opioid use.** Odds ratio (95% CI). Adjusted for age, sex, chronic infections, chronic inflammatory disease, diabetes, cardiac disease, renal disease, end stage renal disease, gout, liver failure, organ transplant, thyroid disease, neurologic disease, (semi)private insurance status, living in an Italian/French speaking canton, managed care model, and pharmacological agents as a proxy for the complexity of the episode.

chronic: OR 0.46, 0.43–0.50). The risk of death was independently association with higher daily doses (50–<100mg: OR 1.19, 1.07–1.22; ≥100mg: OR 1.67, 1.50–1.85). Co-prescribing of benzodiazepine increased the risk of death by 45% (OR 1.45, 1.37–1.53).

## Comorbidities with increased risk for adverse events

Treatments for chronic inflammatory, pulmonary, and psychiatric diseases were consistently associated with an increased risk for all outcomes (S4 Table). The associations between other comorbidities and outcomes were less clear. Treatments for renal and neurologic disease were associated with an increased risk for infections, hospitalizations, and death but not for ED visits. Treatments for cardiac diseases and gout were associated with an increased risk for ED visits, infections, and hospitalization.

## Overall treatment costs per day

We found an independent dose-dependent increase in the overall treatment costs per day. In episodes with ≥100mg MED the treatment costs per day were 34.9% higher compared to episode with a maximum daily dose of <20mg MED (Table 4, Fig 2). Co-prescribing of benzodiazepine increased the treatment costs per day by +4.3%.

## Discussion

The main findings of this study included an increase independent risk for ED visits, hospitalization, and death in patients with higher morphine equivalent dose (MED). The risk was particularly high in episodes with a maximum dose of 100mg MED or more and increased when benzodiazepines were co-prescribed. The chance for ED visits, hospitalization, and antibiotic use increased the longer an episode lasted and was highest in very chronic opioid users (>360 days).

The findings of this study are in line with several previous studies from the UK and the US. Similar to our study, prescription of long-acting opioids for chronic non-cancer pain, compared to anticonvulsants or cyclic antidepressants, was associated with a significantly increased

**Table 4. Overall treatment costs per day in recurrent opioid use.**

| | % increase (95% CI)[†,§] |
|---|---|
| Episode duration (acute reference) | |
| Subacute | -52.3 (-53.7; -50.8) |
| Chronic | -64.4 (-65.1; -63.7) |
| Very chronic | -77.3 (-77.8; -76.7) |
| Maximum daily dose (<20mg reference) | |
| 20 - <50mg | 9.4 (6.5; 12.5) |
| 50 - <100mg | 13.9 (10.9; 17.0) |
| >100mg | 34.9 (31.4; 38.4) |
| Co-prescription | |
| Benzodiazepine | 4.3 (2.6; 6.1) |

[†]Percent increase or decrease treatment costs per day by one unit increase.

[§]Adjusted for age, sex, chronic infections, chronic inflammatory disease, diabetes, cardiac disease, renal disease, end stage renal disease, gout, liver failure, organ transplant, thyroid disease, neurologic disease, (semi)private insurance status, living in an Italian/French speaking canton, managed care model, and pharmacological agents as a proxy for the complexity of the episode.

risk of all-cause mortality [35]. A large cohort study in the primary care setting in the UK showed an increased risk for serious adverse events such as major trauma, addiction and overdose in chronic opioid [25]. The current study revealed in a large sample of patients with recurrent opioid prescriptions for non-cancer related pain treatment, a dose- and time dependent risk for adverse events in opioid use beyond 90 days. Other studies used insurance claims data in the US and found similar associations between opioid prescription and fracture risk [36], risk of overdose [18], and risk of opioid use disorder [37]. Compared to elderly patients

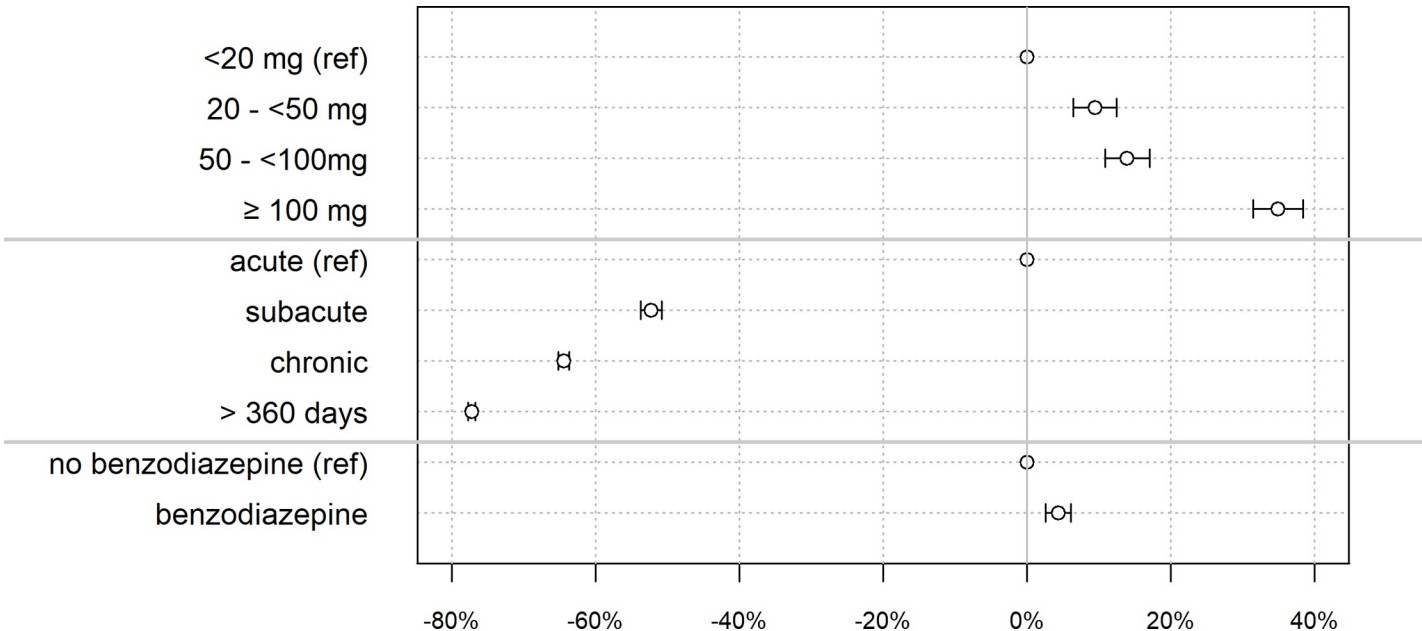

**Fig 2. Increase in average daily treatment costs.** % increase (95% CI). Adjusted for age, sex, chronic infections, chronic inflammatory disease, diabetes, cardiac disease, renal disease, end stage renal disease, gout, liver failure, organ transplant, thyroid disease, neurologic disease, (semi)private insurance status, living in an Italian/French speaking canton, managed care model, and pharmacological agents as a proxy for the complexity of the episode.

without opioid use, opioid prescriptions of ≥50 mg per day resulted in a two-fold increased fracture risk [36]. The risk for overdose was 8.9% higher in patients with a daily dose of 100mg or more MED compared 20mg or less [18]. The risk for unintentional overdose increased with increasing MED dose per day and longer duration [37]. The risk of unintentional overdose was more likely in long acting compared to short-acting opioids [38].

Concomitant benzodiazepine use is discouraged by guidelines [39, 40] due to the increased risk of substance use, greater pain severity, higher rate of mental health conditions, ED visits, and unintentional overdose [41–43]. Our study confirmed that an additional use of benzodiazepines was independently associated with an increased risk for adverse events. We observed in patients with chronic inflammatory, pulmonary, and psychiatric diseases an increased risk for all adverse events. Further, renal and neurologic disease were associated with an increased risk for infections, hospitalizations, and ED visits. Whether this observation indicate that patients with those comorbidities are at an increased risk for adverse events, when opioid treatments are prescribed needs to be further investigated.

### Strength and limitations

The main strength of this study is large sample of opioid episode identified from a representative sample of the Swiss population using insurance claims data. The claims data offers the opportunity to adjust for variables across a wide spectrum of potential confounders. Several limitations need to be discussed. First, insurance claims data does not include the clinical diagnosis and information on disease severity. We tried to mitigate this by adjusting for potential confounders using chronic disease measures based on the medications that are reimbursed. Although we found several comorbidities to be associated with an increased risk for adverse events, we cannot infer causality. Second, the MED dose during the episodes were calculated based on the claims dates and may result in an over- or underestimation of the true dose that was prescribed. Third, we do not know whether patients receiving opioids did also take them. We restricted the analysis to recurrent claims including at least one prescription of a strong opioids. We therefore assume that we excluded patients with singular or very short use of opioids.

### Implication for clinical practice

Chronic opioid use in non-cancer pain should be initiated only if other treatment options fail and short-acting opioids at the lowest dose are recommended [40]. A concomitant prescription of benzodiazepines should be avoided.

### Implication for research

Further prospective studies need to assess the risk of opioid therapy in patients with specific comorbidities. Additionally, prospective studies need to assess modifiable factors that increase the risk for adverse events in patients receiving opioid treatments for non-cancer pain.

### Conclusion

The length of opioid use and higher prescribed morphine equivalent dose were independently associated with an increased risk for ED visits and hospitalizations. The risk for infections, ED visits, hospitalizations, and death also increased with concomitant benzodiazepine use.

## Supporting information

**S1 Table. ATC codes, route of administration and morphine equivalents for opioids.** Adm. R, administration route; O, oral; P, parenteral; R, rectal; SL, sublingual; TD, transdermal; N, nasal; DDD, defined daily dose is the assumed average maintenance dose per day for a drug used for its main indication in adults (29); U, unit; morpheq, Morphine Equivalent Conversion Factor (strength of opioid drug in mg per unit x quantity of units per reimbursed package x number of packages x conversion factor for morphine equivalents. Transmucosal fentanyl conversion MED in milligram for transdermal fentanyl patches were calculated: (mcg/hour (according to the package reimbursed) x 72 hours' x number of patches per package x number of packages reimbursed x 100 [fentanyl conversion factor]) / 1000. MED in milligram for transdermal buprenorphine patches were calculated: (mcg/h according to the package reimbursed x 96 hours' x number of patches per package x number of packages reimbursed x 95 [buprenorphine conversion factor]) / 1000. *All DDD are based on the WHO ATC provided daily dose except for codeine. In Switzerland, codeine is available in combination with paracetamol for pain treatment. No DDD from the WHO were available for codeine-combinations. Therefore, the average treatment dose of the combinations was used to calculate DDD: e.g. Co-Dafalgan⃝R four times daily = 4x20mg codeine.
(DOCX)

**S2 Table. Summary of codes that define cancer related opioid use.**
(DOCX)

**S3 Table. Definitions of comorbidities.**
(DOCX)

**S4 Table. Full models to predict adverse events.**
(DOCX)

## Author Contributions

**Conceptualization:** Jakob M. Burgstaller, Ulrike Held, Andri Signorell, Johann Steurer, Maria M. Wertli.

**Data curation:** Andri Signorell.

**Formal analysis:** Andri Signorell.

**Investigation:** Ulrike Held, Eva Blozik, Maria M. Wertli.

**Methodology:** Jakob M. Burgstaller, Ulrike Held, Andri Signorell, Eva Blozik, Johann Steurer, Maria M. Wertli.

**Project administration:** Maria M. Wertli.

**Resources:** Eva Blozik, Johann Steurer.

**Software:** Andri Signorell.

**Supervision:** Jakob M. Burgstaller, Ulrike Held, Maria M. Wertli.

**Validation:** Jakob M. Burgstaller, Ulrike Held, Andri Signorell, Eva Blozik, Johann Steurer, Maria M. Wertli.

**Writing – original draft:** Maria M. Wertli.

**Writing – review & editing:** Jakob M. Burgstaller, Ulrike Held, Andri Signorell, Eva Blozik, Johann Steurer.

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
