## [Decision Letter · Decision Letter 0]

3 Jan 2020

PONE-D-19-34609

Increased risk of adverse events in non-cancer patients with chronic and high-dose opioid use – a health insurance claims analysis

PLOS ONE

Dear Dr. Wertli,

Thank you for submitting your manuscript to PLOS ONE. After careful consideration, we feel that it has merit but does not fully meet PLOS ONE’s publication criteria as it currently stands. Therefore, we invite you to submit a revised version of the manuscript that addresses the points raised during the review process.

We would appreciate receiving your revised manuscript by Feb 17 2020 11:59PM. To enhance the reproducibility of your results, we recommend that if applicable you deposit your laboratory protocols in protocols.io, where a protocol can be assigned its own identifier (DOI) such that it can be cited independently in the future. For instructions see: http://journals.plos.org/plosone/s/submission-guidelines#loc-laboratory-protocols

We look forward to receiving your revised manuscript.

Kind regards,

Vijayaprakash Suppiah, PhD

Academic Editor

PLOS ONE

Journal Requirements:

2. Please correct your reference to "p=0.000" to "p<0.001" or as similarly appropriate, as p values cannot equal zero.

"MW, JB, JS, and UH have no competing interest to declare. AS and EB are employed by the Helsana Group. The sponsor had no role in the planning, conducting or submission of this manuscript. These authors declare no conflict of interest. Helsana Group shall have no liability to any third party in respect to the contents of this article. All the other authors have no conflict of interests or financial disclosures to declare."

5. Your ethics statement must appear in the Methods section of your manuscript. If your ethics statement is written in any section besides the Methods, please move it to the Methods section and delete it from any other section. Please also ensure that your ethics statement is included in your manuscript, as the ethics section of your online submission will not be published alongside your manuscript.

6. Please include your tables as part of your main manuscript and remove the individual files. ** Please note that supplementary tables (should remain/ be uploaded) as separate "supporting information" files **

Reviewers' comments:

Reviewer's Responses to Questions

**Comments to the Author**

1. Is the manuscript technically sound, and do the data support the conclusions?

Reviewer #1: Yes

Reviewer #2: Partly

2. Has the statistical analysis been performed appropriately and rigorously? 

Reviewer #1: Yes

Reviewer #2: Yes

3. Have the authors made all data underlying the findings in their manuscript fully available?

Reviewer #1: No

Reviewer #2: No

4. Is the manuscript presented in an intelligible fashion and written in standard English?

Reviewer #1: Yes

Reviewer #2: Yes

5. Review Comments to the Author

Reviewer #1: This is an interesting and important paper but with one challenging statistical design issue. It appeared that longer duration (e.g., more than 360 days of opioid use) generally leads to lower risk of death, but one needs to be alive to complete 360 days of opioid use -- thus there is most likely a mechanical association between opioid use duration and mortality.

On page 13, line 4: "The majority of patients, died during an acute episode" I would write it differently, that the majority of deaths occurred during an acute episode.

Finally, I would have liked to see more results using actual percentages related to adverse outcomes, rather than odds ratios, for two reasons. First, knowing the likelihood of adverse events is always helpful to get a context of how important the problem is, and second, odds ratios have well-known interpretation issues when the likelihood of the adverse event is high. It may not be much of a problem here, but I would like to see something there.

Reviewer #2: In this study a large Swiss insurance database was used to estimate adverse events associated with different levels and durations of opioid prescribing. The outcomes are largely in line with those of previous studies although the Swiss population perhaps has not been the focus of previous analyses.

1. The WHO ladder was not intended nor validated to be used for non-cancer pain. Certainly, there has been a massive increase in opioid prescribing in many countries, but it is not clear that mistaken adherence to a now very old cancer-related algorithm is much to blame. Aren’t there much stronger reasons like pharma company promotion, efforts of advocacy groups and strongly expressed although poorly justified opinions of “experts?” Perhaps these should be included in the first paragraph.

2. The description of medical benefits is appreciated. However, some clear statement of the lowlihood of capturing most or all opioid prescriptions for the cohort in the database would be helpful.

3. Episodes occurring within the first week after prescribing were excluded because they were thought to be likely related directly to the opioid. These may in fact be some of the most interesting occurrences! It would be OK to analyze these separately, but please do include them along with prescribed dose relationships. Also, it was confusing why a 6-day period of exclusion for outcome was described in one section of the methods and a 2 week period seemed to be described in the “Outcome of interest” section.

4. Is there validation or even rationale for using the identity of the pharmacological agent as a proxy for the complexity of the episode? Why does this matter and couldn’t choice be dues to the arbitrary practices of individual providers?

5. Please clarify whether opioid overdose cold have been included as an adverse effect, or even a diagnosis related to substance misuse These would be much easier to interpret.

6. Acute opioid prescribing seems to have been used as the reference. Why not a propensity-matched no-opioid group? Unless there is a very compelling reasons not to do this, a no-opioid matched group should be included.

7. Tables are very hard to read by themselves. Please make better use of graphical representations of key data.

6. PLOS authors have the option to publish the peer review history of their article (what does this mean?). If published, this will include your full peer review and any attached files.

Reviewer #1: No

Reviewer #2: No

---

## [Decision Letter · Decision Letter 1]

29 Apr 2020

PONE-D-19-34609R1

Increased risk of adverse events in non-cancer patients with chronic and high-dose opioid use – a health insurance claims analysis

PLOS ONE

Dear Dr. Wertli,

Thank you for submitting your manuscript to PLOS ONE. After careful consideration, we feel that it has merit but does not fully meet PLOS ONE’s publication criteria as it currently stands. Therefore, we invite you to submit a revised version of the manuscript that addresses the points raised during the review process.

We would appreciate receiving your revised manuscript by Jun 13 2020 11:59PM. To enhance the reproducibility of your results, we recommend that if applicable you deposit your laboratory protocols in protocols.io, where a protocol can be assigned its own identifier (DOI) such that it can be cited independently in the future. For instructions see: http://journals.plos.org/plosone/s/submission-guidelines#loc-laboratory-protocols

We look forward to receiving your revised manuscript.

Kind regards,

Vijayaprakash Suppiah, PhD

Academic Editor

PLOS ONE

Reviewers' comments:

Reviewer's Responses to Questions

**Comments to the Author**

1. If the authors have adequately addressed your comments raised in a previous round of review and you feel that this manuscript is now acceptable for publication, you may indicate that here to bypass the “Comments to the Author” section, enter your conflict of interest statement in the “Confidential to Editor” section, and submit your "Accept" recommendation.

Reviewer #3: All comments have been addressed

Reviewer #4: (No Response)

2. Is the manuscript technically sound, and do the data support the conclusions?

Reviewer #3: Yes

Reviewer #4: Yes

3. Has the statistical analysis been performed appropriately and rigorously? 

Reviewer #3: Yes

Reviewer #4: Yes

4. Have the authors made all data underlying the findings in their manuscript fully available?

Reviewer #3: Yes

Reviewer #4: Yes

5. Is the manuscript presented in an intelligible fashion and written in standard English?

Reviewer #3: Yes

Reviewer #4: Yes

6. Review Comments to the Author

Reviewer #3: (No Response)

Reviewer #4: 1. You exclude cancer-related opioid treatment. Are there other conditions that you found were treated with opioids at a high rate?

2. Are there combinations of medical conditions that also lead to high opioid use and treatment? That combination of conditions could also contribute to adverse medical outcomes. Correspondingly, a recommendation could be that for a patient with this set of conditions faces more adverse medical outcomes from strong opioid treatment.

7. PLOS authors have the option to publish the peer review history of their article (what does this mean?). If published, this will include your full peer review and any attached files.

Reviewer #3: Yes: Terri Voepel-Lewis

Reviewer #4: No

---

## [Author Response · Author response to Decision Letter 1]

12 May 2020

See attached Response to the Reviewer.

---

## [Decision Letter · Decision Letter 2]

14 Aug 2020

Increased risk of adverse events in non-cancer patients with chronic and high-dose opioid use – a health insurance claims analysis

PONE-D-19-34609R2

Dear Dr. Wertli,

We’re pleased to inform you that your manuscript has been judged scientifically suitable for publication and will be formally accepted for publication once it meets all outstanding technical requirements.

Kind regards,

Vijayaprakash Suppiah, PhD

Academic Editor

PLOS ONE

Additional Editor Comments (optional):

The authors have sufficiently addressed the comments and concerns raised by the previous reviewers. I recommend that this manuscript be accepted without further amendments. 

Reviewers' comments:

Reviewer's Responses to Questions

**Comments to the Author**

1. If the authors have adequately addressed your comments raised in a previous round of review and you feel that this manuscript is now acceptable for publication, you may indicate that here to bypass the “Comments to the Author” section, enter your conflict of interest statement in the “Confidential to Editor” section, and submit your "Accept" recommendation.

Reviewer #2: All comments have been addressed

Reviewer #5: (No Response)

2. Is the manuscript technically sound, and do the data support the conclusions?

Reviewer #2: Yes

Reviewer #5: Partly

3. Has the statistical analysis been performed appropriately and rigorously? 

Reviewer #2: I Don't Know

Reviewer #5: No

4. Have the authors made all data underlying the findings in their manuscript fully available?

Reviewer #2: Yes

Reviewer #5: (No Response)

5. Is the manuscript presented in an intelligible fashion and written in standard English?

Reviewer #2: Yes

Reviewer #5: Yes

6. Review Comments to the Author

Reviewer #2: The final analysis is consistent with other reports, but extends the findings to another wealthy Western nation. The death rate from opioid use is perhaps even more dramatic than in some other countries.

Reviewer #5: The motivation of the study is important, i.e., assessing the risk of adverse events in recurrent opiod users for non-cancer pain in a Swiss population. I have some questions on the statistical analysis conducted.

1. Although the outcome variables were clearly specified, it is hard to understand their statistical nature. Please specify them as discrete, or binary.

2. In "Statistical Methods" section, the authors utilized logistic regression on the binary outcomes, and stated "....including disease duration (....), maximum prescribed dose (....)...". This makes a reader confused; please state that these are covariates you are considering.

3. You are analyzing via R, and likely, the 4 binary responses (corresponding to a subject) can be correlated. It is likely that emergency visits may lead to hospitalization, infections, and other events. The regression analysis can be strengthened by producing odds ratios (or something similar), where information from these 4 variables can be combined. There are several methods available; I leave it upon the authors to choose one that best suits their specific example.

7. PLOS authors have the option to publish the peer review history of their article (what does this mean?). If published, this will include your full peer review and any attached files.

Reviewer #2: No

Reviewer #5: No

---

## [Editor Report · Acceptance letter]

3 Sep 2020

PONE-D-19-34609R2 

Increased risk of adverse events in non-cancer patients with chronic and high-dose opioid use – a health insurance claims analysis 

Dear Dr. Wertli:

I'm pleased to inform you that your manuscript has been deemed suitable for publication in PLOS ONE. Congratulations! Your manuscript is now with our production department. 

Kind regards, 

on behalf of

Dr. Vijayaprakash Suppiah 

Academic Editor

PLOS ONE